# Retinal Biomarkers of Cerebral Small Vessel Disease: A Systematic Review

**Elena Biffi** [1]*, **Zachary Turple**[1], **Jessica Chung**[1], **Alessandro Biffi**[2,3,4]

**1** New England College of Optometry, Boston, MA, United States of America, **2** Henry and Allison McCance Center for Brain Health, Massachusetts General Hospital, Boston, MA, United States of America, **3** Department of Neurology, Massachusetts General Hospital, Boston, MA, United States of America, **4** Department of Psychiatry, Massachusetts General Hospital, Boston, MA, United States of America

* BiffiE@neco.edu

## Abstract

### Introduction

Cerebral Small Vessel Disease (CSVD), a progressive degenerative disorder of small caliber cerebral vessels, represents a major contributor to stroke and vascular dementia incidence worldwide. We sought to conduct a systematic review of the role of retinal biomarkers in diagnosis and characterization of CSVD.

### Methods

We conducted a systematic review of MEDLINE, PubMed, Scopus, the Cochrane Library Database, and Web of Science. We identified studies of sporadic CSVD (including CSVD not otherwise specified, Cerebral Amyloid Angiopathy, and Hypertensive Arteriopathy) and the most common familial CSVD disorders (including CADASIL, Fabry disease, and MELAS). Included studies used one or more of the following tools: visual fields assessment, fundus photography, Optical Coherence Tomography and OCT Angiography, Fluorescein Angiography, Electroretinography, and Visual Evoked Potentials.

### Results

We identified 48 studies of retinal biomarkers in CSVD, including 9147 cases and 12276 controls. Abnormalities in retinal vessel diameter (11 reports, n = 11391 participants), increased retinal vessel tortuosity (11 reports, n = 617 participants), decreased vessel fractal dimension (5 reports, n = 1597 participants) and decreased retinal nerve fiber layer thickness (5 reports, n = 4509 participants) were the biomarkers most frequently associated with CSVD. We identified no reports conducting longitudinal retinal evaluations of CSVD, or systematically evaluating diagnostic performance.

### Conclusion

Multiple retinal biomarkers were associated with CSVD or its validated neuroimaging biomarkers. However, existing evidence is limited by several shortcomings, chiefly small sample size and unstandardized approaches to both biomarkers' capture and CSVD

**Data Availability Statement:** All relevant data are within the paper and its Supporting Information files.

**Funding:** This study was supported by NIH T35EY007149, and by grants from the National

Academy of Medicine and the American Optometric Association. The funding entities played no role in the study design, data collection and analysis, decision to publish, or preparation of the manuscript.

**Competing interests:** No conflicts of interest to disclose.

characterization. Additional larger studies will be required to definitively determine whether retinal biomarkers could be successfully incorporated in future research efforts and clinical practice.

## Introduction

Cerebral Small Vessel Disease (CSVD) is a progressive, age-related degenerative disorder of the small caliber vessels of the Central Nervous System (CNS) [1–3]. Due to progressive accumulation of microvascular lesions over time, it is responsible for almost 20% of ischemic stroke and over 80% of all hemorrhagic stroke [4]. CSVD also represents the second most common form of dementia, following Alzheimer's disease [1,5]. The vast majority of CSVD cases are sporadic in nature, presenting without a clear familial inheritance pattern (Table 1).

Most sporadic CSVD cases (over 90% of all CSVD diagnoses) are accounted for by two progressive, aging-related disorders: Cerebral Amyloid Angiopathy (CAA) and Hypertensive Arteriopathy (HTNA) [1]. Rare familial forms occurring on a hereditary basis (usually monogenic autosomal dominant) have also been identified, with the most frequently reported being Cerebral Autosomal Dominant Arteriopathy with Subcortical Infarcts and Leukoencephalopathy (CADASIL), Fabry disease and Mitochondrial Encephalomyopathy, Lactic Acidosis and Stroke-like episodes (MELAS) syndrome [1,6]. Other rarer subtypes of CSVD include infectious and immune-mediate forms (Table 1).

**Table 1. Cerebral small vessel disease subtypes.**

| Sporadic CSVD | Familial (Hereditary) CSVD |
|---|---|
| Age-related | CADASIL |
| • CAA | CARASIL |
| • HTNA | MELAS |
| Immune-mediated | Fabry Disease |
| • Primary CNS Vasculitis | CSVD due to Type IV Collagen Disease |
| • Secondary CNS Vasculitis | Retinal Vasculopathy with Cerebral Leukoencephalopathy |
| • ANCA-associated vasculitis | Hereditary CAA |
| • Hypersensitivity vasculitis | • Dutch Variant Hereditary CAA |
| • CNS Vasculitis due to SLE | • Flemish Variant Hereditary CAA |
| • CNS Vasculitis due to Sjogren | • Italian Variant Hereditary CAA |
| • Rheumatoid Vasculitis | • Piedmont Variant Hereditary CAA |
| • CNS Vasculitis due to MCTD | • Arctic Variant Hereditary CAA |
| • CNS Vasculitis due to Behçet | • Icelandic Variant Hereditary CAA |
| Infectious | • Iowa Variant Hereditary CAA |
| • HIV CNS Vasculitis | • Meningovascular Amyloidosis |
| • Meningovascular Syphilis | |
| • CMV Vasculitis | |
| • VZV Vasculitis | |
| • HBV and HCV Vasculitis | |
| • Cerebral malaria | |

Abbreviations: CAA = Cerebral Amyloid Angiopathy, CADASIL = Cerebral Autosomal Dominant Arteriopathy with Subcortical Infarcts and Leukoencephalopathy, CARASIL = Cerebral Autosomal Recessive Arteriopathy with Subcortical Infarcts and Leukoencephalopathy, CNS = Central Nervous System, CSVD = Cerebral Small Vessel Disease, HTNA = Hypertensive Arteriopathy, MCTD = Mixed Connective Tissues Disease, MELAS = Mitochondrial Encephalomyopathy, Lactic Acidosis and Stroke-like episodes, SLE = Systemic Lupus Erythematosus.

While definitive diagnosis of sporadic and familial forms of CSVD requires histopathological examination of brain tissue, in clinical practice the diagnostic gold standard is identification of typical CSVD-related ischemic and hemorrhagic lesions on MRI brain imaging. The neuroimaging biomarkers most commonly associated with CSVD include white matter hyperintensities (also referred to as leukoaraiosis), lacunar infarcts, dilated perivascular spaces, cortical superficial siderosis, and cerebral microbleeds [7]. However, these findings are consistent with the presence of irreversible ischemic or hemorrhagic CNS damage, and are, therefore, of limited use in the diagnosis and monitoring of the preclinical and minimally symptomatic stages of CSVD [2]. In addition, financial (scan and personnel costs) and logistical (availability of equipment and expertise) limitations prevent the widespread use of MRI neuroimaging in early screening for CSVD and monitoring of disease progression and response to treatment over time [1].

The retina contains CNS neurons and a small vessel network displaying close anatomical and physiological parallels with the corresponding cerebral neurovascular unit [8]. It is, therefore, conceivable that non-invasive evaluation of retinal neurons and vessels may provide novel biomarkers for CSVD diagnosis and staging [9,10]. Because retinal biomarkers provide information on tissue structure and function at the microscopic level, they may allow for diagnosis of CSVD in earlier, less symptomatic or asymptomatic stages. Finally, retinal biomarkers compare very favorably with MRI-based neuroimaging in terms of equipment availability, operating costs, and expertise required to gather data [8]. Therefore, they may allow for large-scale screening for CSVD in at-risk population (e.g. elderly individuals), in a way MRI neuroimaging cannot due to prohibitive costs and insufficient number of scanners and trained personnel available.

To date, several studies have tested this overall hypothesis using a variety of different technologies, including visual fields (VF) assessment, fundus photography, Optical Coherence Tomography (OCT) and OCT Angiography (OCTA), Fluorescein Angiography (FA), Electroretinography (ERG), and Visual Evoked Potentials (VEP) [9,10]. All these biomarker acquisition modalities offer a variety of potential advantages over MRI neuroimaging, including widespread availability as part of routine medical care and ability to evaluate neurons and blood vessels at the microscopic level (which is currently possible only in a very limited fashion with MRI neuroimaging) [8].

To date, no retinal biomarkers have emerged as candidates for adoption into routine diagnostic or clinical care practice for CSVD. The present systematic review aims to evaluate existing evidence on the performance of retinal biomarkers in the diagnosis and staging of different forms of CSVD. Our primary goal is to identify retinal biomarkers demonstrating associations with: 1) CSVD diagnoses (in affected individuals vs. healthy controls); 2) established neuroimaging markers of CSVD; 3) acute stroke risk or cognitive decline secondary to CSVD. We also sought to identify studies reporting diagnostic performance for different CSVD disorders, whether in initial screening or longitudinal monitoring. Finally, we evaluated the strengths and gaps in currently available evidence on a disease and technology-specific basis in order to better inform future research efforts.

## Material and methods

### Review rationale and overall design

This systematic review was conducted on the basis of a pre-specified protocol and designed in agreement with the Preferred Reporting Items for Systematic Reviews and Meta-Analyses (PRISMA) 2020 guidelines [11]. We chose to focus on studies of CSVD, in both its sporadic (CSVD not otherwise specified, CAA, or HTNA) and most common familial (CADASIL,

Fabry disease, and MELAS) forms [1,6]. The inclusion of MELAS disease (primarily a mitochondrial disease) in the present analyses is motivated by findings indicating small vessel vasculopathy secondary to energy failure as central to the characteristic ischemic events in this condition [12]. For sporadic CSVD forms, we focused on studies utilizing established neuropathological or neuroimaging criteria for diagnosis [1,5]. For familial CSVD forms, we focused on studies with confirmed genetic diagnoses and consistent clinical and neuroimaging phenotypes [6].

We pre-specified inclusion of the following retinal evaluation modalities: visual fields (VF) assessment, fundus photography, Optical Coherence Tomography (OCT) and OCT Angiography (OCTA), Fluorescein Angiography (FA), Electroretinography (ERG), and Visual Evoked Potentials (VEP) [13]. Our primary pre-specified objective was to identify retinal biomarkers distinguishing CSVD cases from controls. As a secondary objective, we sought to determine whether retinal biomarkers (individually or in combination) were found to be associated with either: 1) neuroimaging markers of CSVD severity; or 2) clinical metrics of acute stroke risk or cognitive decline secondary to CSVD. We defined CSVD-related MRI markers according to the Standards for Reporting Vascular Changes on Neuroimaging (STRIVE) guidelines [7]. All analyses were conducted using publicly available summary data, without any access to individual level data. As such, no institutional review board approval or informed patient consent was required.

## Search strategy

We conducted an online literature search using the following publicly accessible databases: Medical Literature Analysis and Retrieval System Online (MEDLINE), PubMed, Scopus, the Cochrane Library Database, and Web of Science. Please refer to Supporting Information (S1 File) for details on the Search Strategy. We restricted our search to studies published in English. Following initial database queries, results were harmonized in a single list of publications. We then manually reviewed references of relevant articles to identify additional potentially relevant publications via forward citation search. After completing this step, the initial publication list was pruned from duplicate entries (Fig 1). We then reviewed study abstracts to identify studies that met the following criteria: 1) included original data from human participants; 2) investigated one or more retinal biomarkers generated using the pre-specified methodologies; 3) compare the distribution of retinal biomarkers across CSVD patients, between CSVD patients and healthy controls, across patient groups identified by established CSVD neuroimaging markers, or across patient groups identified by stroke risk and/or cognitive performance measures. In order to qualify for a diagnosis of CSVD, participants in a study had to be present with either: 1) CSVD-related lacunar ischemic stroke; 2) CSVD-related spontaneous intracerebral hemorrhage; 3) CSVD-related cognitive decline fulfilling criteria for Vascular contributions to Cognitive Impairment and Dementia (VCID). We specifically excluded studies that did not confirm that stroke or cognitive decline were attributable to CSVD based on current diagnostic criteria [14,15]. We specifically excluded the following article types: 1) review studies; 2) individual case studies; 3) study protocols; 4) conference presentations, abstracts, or summaries; 5) comments on original research that did not present novel peer-review findings; 6) editorial commentaries, viewpoints, and other opinion pieces. For previously published systematic meta-analyses, we separately evaluated each included study (if not already identified as part of our search strategy) for inclusion in our systematic review. When a determination about meeting inclusion and exclusion criteria could not be reached via abstract review, studies were marked for full-text review.

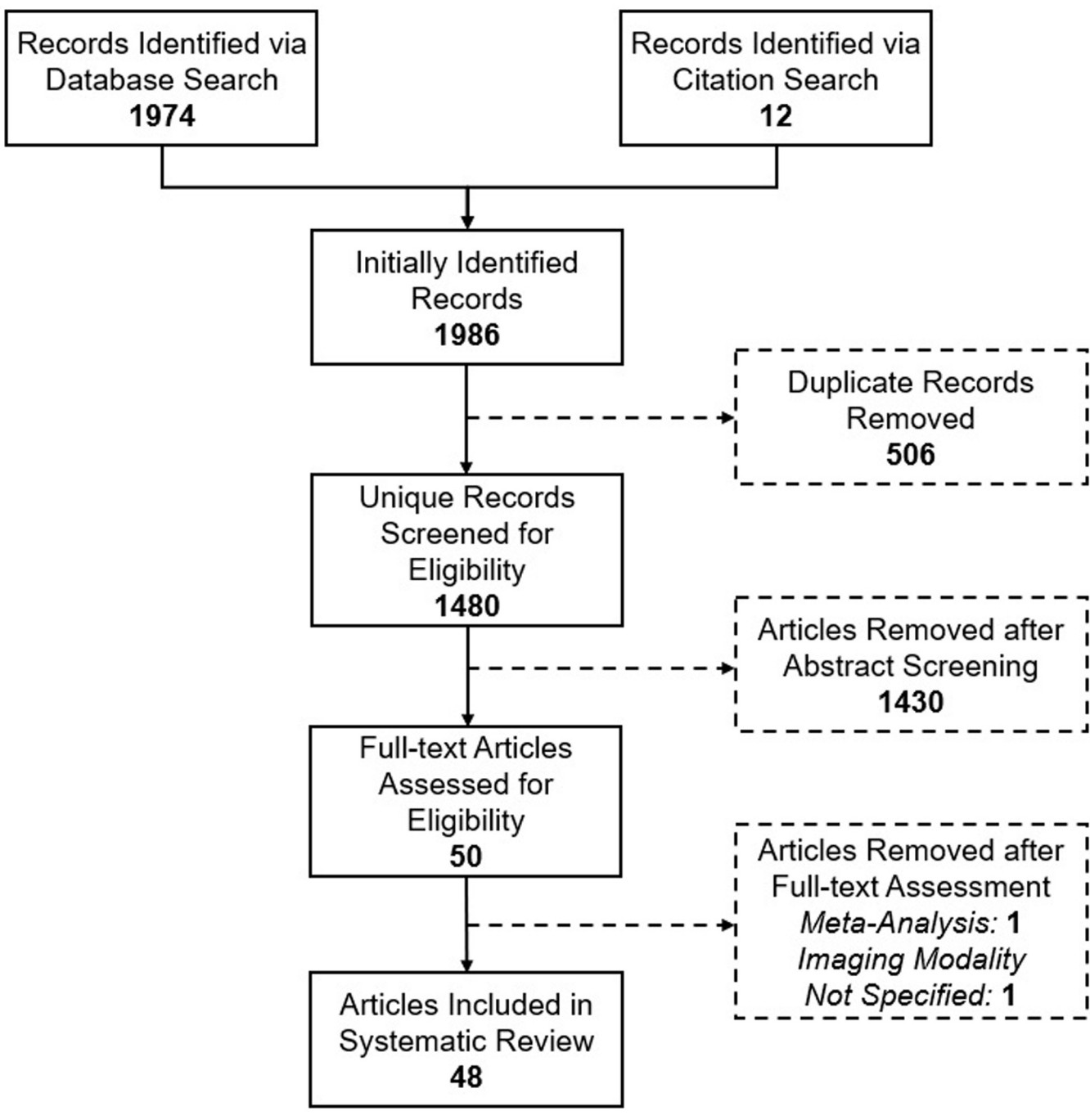

**Fig 1. Search strategy flow chart.**

## Data extraction

Following initial screening of potentially relevant publications, eligibility for inclusion in the present review was confirmed via full-text review. We pre-specified for extraction, from each individual article, the following data points: authors, publication year, pre-specified study aim / hypotheses, study type, number of patients and controls, participants' demographics (number of male vs. female, mean age), participant selection criteria, CSVD diagnostic criteria

employed, MRI neuroimaging (if applicable), cognitive performance evaluation (if applicable), genetic diagnostic testing (if applicable), retinal evaluation modality, device and imaging settings, image quality control procedures, retinal biomarkers extracted and extraction methodology, outcomes of interest, and statistical modeling methods. Data extraction was conducted by two separate reviewers (ZT and JC) independently and blinded to each other. All extracted data points were cross-checked, and disagreements reconciled via joint evaluation by a board-certified optometrist with expertise in ocular imaging (EZB) and a board-certified neurologist with expertise in CSVD (AB).

## Study quality assessment

We performed systematic assessment of study quality for eligible publications in agreement with the STrengthening the Reporting of OBservational studies in Epidemiology (STROBE) recommendations as qualifying items [16]. We used the STROBE checklist to asses study quality based on whether or not individual recommendation items were successfully addressed, with a final score ranging from 0 (none of the recommendations addressed) to 22 (all recommendations addressed). We separately scored studies of OCT and OCTA markers in CSVD using the Advised Protocol for OCT Study Terminology and Elements (APOSTEL) v2.0 recommendations, which provide an optimal framework for design, execution, and reporting of results in quantitative OCT/OCTA studies [17]. Using an identical procedure as for the STROBE study quality score, we assigned individual publications values ranging from 0 (none of the recommendations addressed) to 9 (all recommendations addressed). We did not identify specific recommendations for other retinal evaluation methodologies that could be applied to evaluate study quality. Of note, study quality was evaluated after the final list of included publications was generated and, therefore, had no impact on whether individual articles were included or excluded from the present analyses.

## Data analysis

Based on prior reviews on similar topics, we expected to identify a small number of studies investigating each individual form of CSVD with a specific retinal evaluation modality [8,9,18–20]. In addition, we expected included studies to report on a variety of retinal biomarkers with widely differing definitions. We, therefore, did not pre-specify methods for meta-analysis of published evidence, but rather opted to focus on a systematic presentation of results. We chose to collate all associations between retinal biomarkers and CSVD disorders, subdivided by disease of interest and data acquisition modality.

## Results

### Search results

Our initial automated searches of online repositories identified a total of 1974 reports fitting the search criteria. We identified an additional 12 reports via manual examination and automated cross-reference of citations from publications identified via our search strategy. After elimination of 506 duplicated records, we screened for eligibility 1480 publications (Fig 1). A total of 1430 reports were excluded after manual review of abstracts for failing to satisfy all inclusion and exclusion criteria. We therefore conducted full-text manual review of 50 papers. Among these, one was excluded as it presented a meta-analysis of previously published primary data. As per our pre-specified methodology, we included all meta-analyzed studies (if they individually met our eligibility criteria) in our review. Another publication was excluded

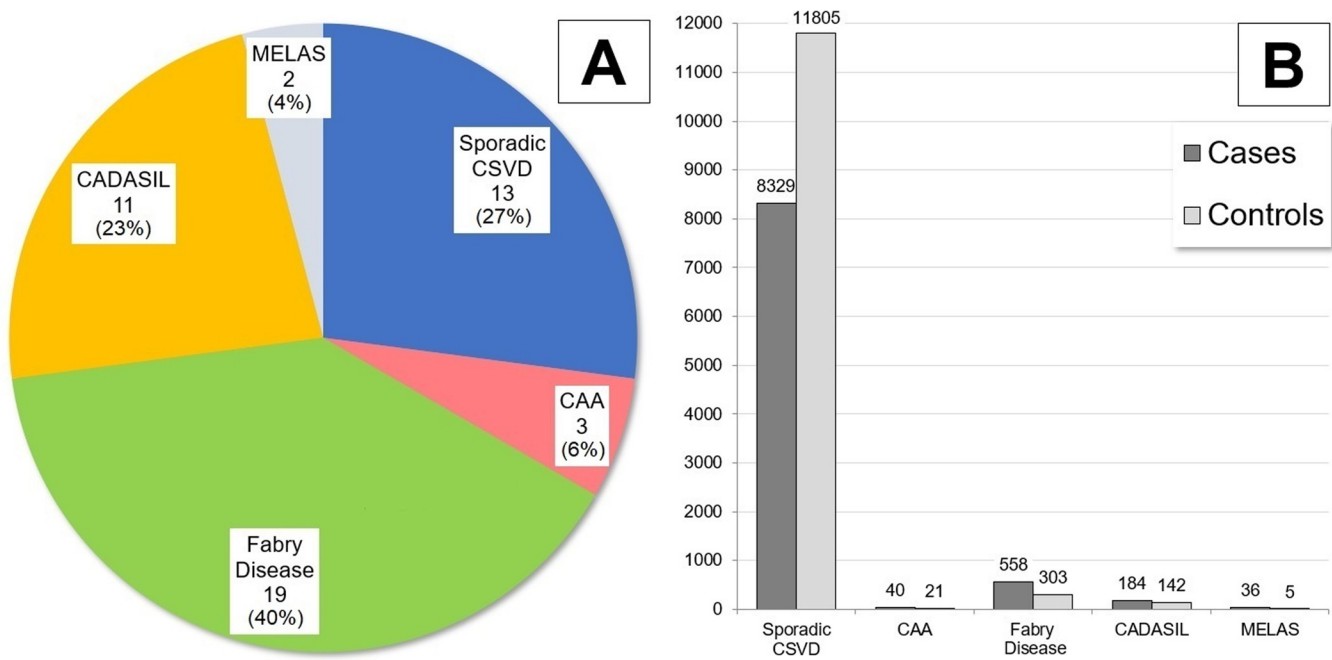

**Fig 2. Number and sample size of studies included in systematic review.** Panel A: Number and percentages of studies included in the present systematic review, based on CSVD disorder of interest. Panel B: Number of affected (cases) and healthy (controls) individuals participating in studies included in the present systematic review, based on CSVD disorder of interest. Abbreviations: CAA = Cerebral Amyloid Angiopathy, CADASIL = Cerebral Autosomal Dominant Arteriopathy with Subcortical Infarcts and Leukoencephalopathy, CSVD = Cerebral Small Vessel Disease, MELAS = Mitochondrial Encephalomyopathy, Lactic Acidosis, and Stroke-like episodes syndrome.

as the retinal evaluation modality employed could not be definitively ascertained. We therefore included 48 separate published reports of studies of retinal biomarkers in CSVD (Fig 1).

## Studies included in systematic review

We present information on the number and percentage of studies focusing on CSVD in general (henceforth referred to as "sporadic", to distinguish from familial monogenic forms), CAA, CADASIL, Fabry disease and MELAS in Fig 2 (Panel A).

It is worth nothing that while three papers specifically applied diagnostic criteria to enroll patients with CAA, none of the remaining papers focused on sporadic CSVD applied criteria specific to either CAA or HTNA. Rather, these studies defined participants on the basis of clinical presentation and neuroimaging as being diagnosed with CVSD (or similar terminology), without further specification. As visually illustrated in Fig 2 (Panel B), individuals enrolled in these studies of sporadic CSVD represented the overwhelming majority among participants included in the present systematic review, since they accounted for 8329 of 9147 cases (91%), and 11805 of 12276 control (96%).

We present in Table 2 information on retinal evaluation modalities employed by studies included in our systematic review, based on the CSVD disorder of interest.

The vast majority of studies employed a single imaging modality, with only a handful incorporating multiple techniques, and none including all those considered for inclusion in our systematic review. Fig 3 provides a summary of the findings from our systematic review in terms of retinal biomarkers identified.

Vascular retinal biomarkers represented the largest number of studies reporting positive associations, especially for: 1) changes in diameter of retinal vessels (11 reported associations

**Table 2. Summary of imaging modalities and CSVD disorders for studies included in systematic review.**

| | | CSVD Disorders | | | | |
|---|---|---|---|---|---|---|
| | | **Sporadic CSVD** | **CAA** | **Fabry Disease** | **CADASIL** | **MELAS** |
| *Imaging Modalities* | **Fundus Photography** | **10** Ref: [21–30] | **3** Ref: [31–33] | **11** Ref: [34–44] | **5** Ref: [45–49] | **1** Ref: [50] |
| | **OCT** | **2** Ref: [51,52] | **2** Ref: [31,33] | **5** Ref: [34,37–39,53] | **6** Ref: [45,48,54–57] | **1** Ref: [58] |
| | **OCTA** | **2** Ref: [51,59] | **1** Ref: [31] | **8** Ref: [37,44,53,60–64] | **1** Ref: [55] | - |
| | **FA** | - | **1** Ref: [32] | - | **3** Ref: [45,47,48] | - |
| | **ERG** | - | - | **1** Ref: [37] | **2** Ref: [65,66] | **1** Ref: [50] |
| | **VEP** | - | - | - | **2** Ref: [48,57] | - |
| | **VF Assessment** | - | - | **3** Ref: [36,67,68] | **2** Ref: [47,48] | - |

Table presents the number of studies employing specific imaging modalities in each CSVD disorder of interest for the present systematic review. Several studies employed multiple imaging modalities, please refer to the Results section for details. Abbreviations: CAA = Cerebral Amyloid Angiopathy, CADASIL = Cerebral Autosomal Dominant Arteriopathy with Subcortical Infarcts and Leukoencephalopathy, CSVD = Cerebral Small Vessel Disease, ERG = Electroretinography, FA = Fluorescein Angiography, MELAS = Mitochondrial Encephalomyopathy, Lactic Acidosis, and Stroke-like episodes syndrome, OCT = Optical Coherence Tomography, OCTA = Optical Coherence Tomography Angiography, VEP = Visual Evoked Potentials, VF = Visual Field.

among 11391 participants); 2) increased retinal vessel tortuosity (11 reported associations among 617 participants) and decreased vessel fractal dimension (5 reported associations among 1597 participants), both established markers of progressive chronic retinal angiopathy [20]. Among neuronal retinal biomarkers, decreased retinal nerve fiber layer (RNFL) thickness was the only one to be associated with multiple CSVD disorders (5 reported associations among 4509 participants).

We initially planned to compare results on retinal biomarkers across different forms of CSVD to determine whether consistent association patterns emerged, potentially indicating shared pathophysiological mechanisms. However, we found no biomarkers displaying consistent associations across all or even most CSVD disorders of interest. Based on findings from Table 2 and Fig 3, this observation most likely reflects limited overlap in choice of retinal evaluation technologies and specific biomarkers across different studies, rather than underlying biological heterogeneity.

## Study quality assessment

The median quality score based on STROBE recommendations [16] for included studies was 15/22, with inter-quartile range of 11/22 to 19/22. Most studies lost points for failing to appropriately describe study design in title or abstract; failing to explain rationale for study sample size; inadequate explanations provided regarding controlling for potential sources of bias; and inadequate discussion of the generalizability of results. Among 16 studies presenting results of retinal OCT-based imaging in CSVD patients, the median quality score based on APOSTEL v2.0 recommendations [17] was 4/9, with inter-quartile range of 2/9 to 6/9. Most studies lost points for failing to clearly document scanning protocol; acquisition devices (either hardware and/or software); and acquisition settings. Overall, our study quality assessment did raise concerns about a substantial proportion of studies failing to provide detailed information on key aspects of study design (especially sample size and projected power) and study execution, primarily in terms of details pertaining to hardware, software, and parameters used for data acquisition.

## Retinal biomarkers in sporadic CSVD

We identified a total of 13 studies investigating the association between retinal biomarkers and sporadic CSVD (Table 3). These studies included a total of 8329 sporadic CSVD patients and

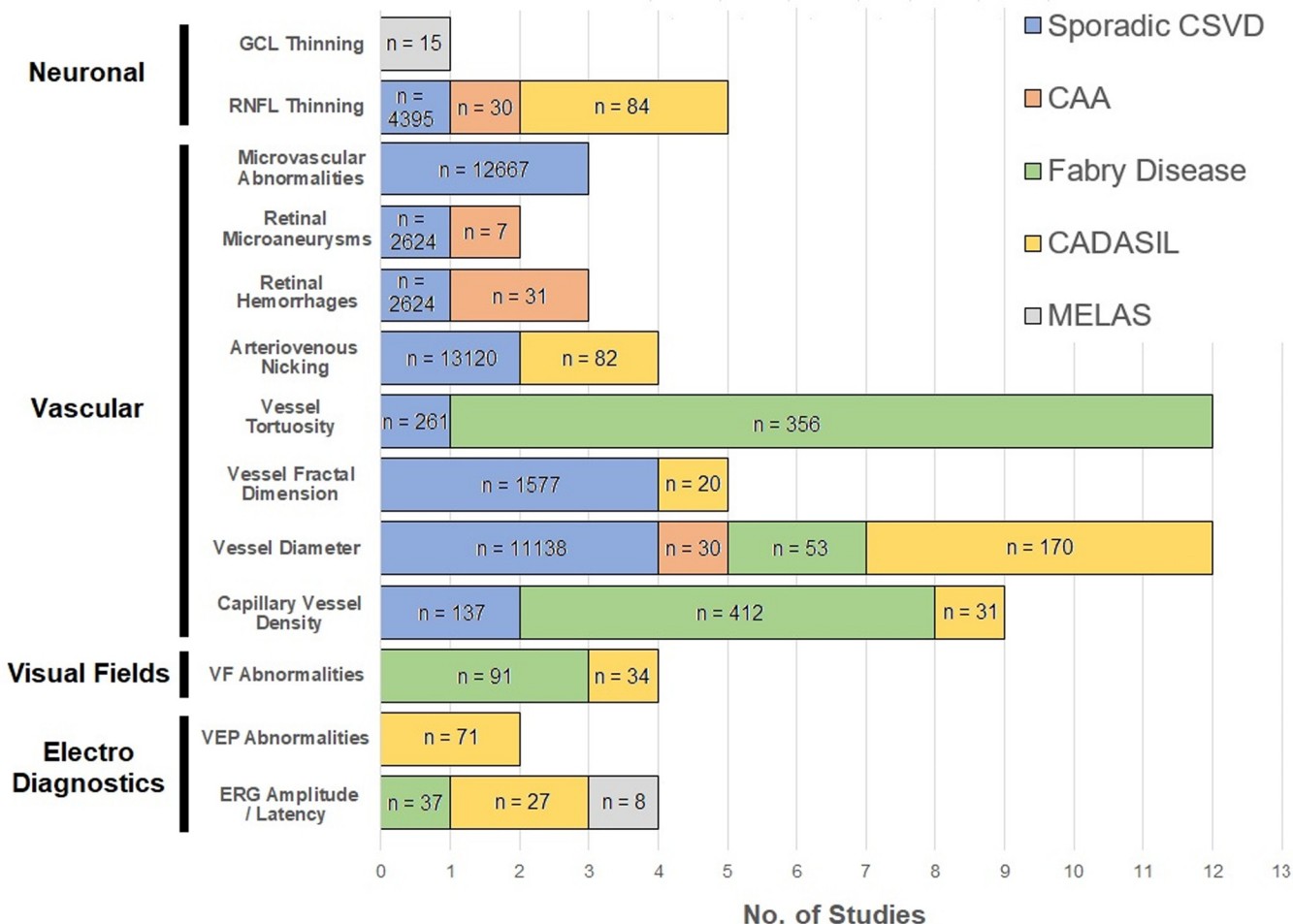

**Fig 3. Retinal biomarkers identified in systematic review.** Figure presents the number of studies identifying specific retinal biomarkers as associated with each CSVD disorder of interest for the present systematic review. For each CSVD disorder, the number of individuals included in studies reporting association of a specific biomarker is reported (denoted as n). Abbreviations: CAA = Cerebral Amyloid Angiopathy, CADASIL = Cerebral Autosomal Dominant Arteriopathy with Subcortical Infarcts and Leukoencephalopathy, CSVD = Cerebral Small Vessel Disease, ERG = Electroretinography, GCL = Ganglion Cell Layer, MELAS = Mitochondrial Encephalomyopathy, Lactic Acidosis, and Stroke-like episodes syndrome, RNFL = Retinal Nerve Fiber Layer, VEP = Visual Evoked Potentials, VF = Visual Field.

11805 controls. The median number of CSVD patients per study was 262 (range 24–4395) and the median number of controls per study was 814 (range 20–10158). We identified 10 reports using fundus photography [21–30], one study using OCT [52], one using OCTA [59], and one combining OCT and OCTA [51]. We found no report leveraging VF assessment, FA or ERG to investigate sporadic CSVD. A total of 4 of 13 studies (31%) utilized ischemic stroke as diagnostic criterion for sporadic CSVD. Evaluation of one or more CSVD neuroimaging biomarkers was included in 7 of 13 studies (54%). Only three studies (21%) utilized vascular cognitive impairment as eligibility criterion. The majority of publications (10 of 13, 77%) provided full details of imaging device and methodology. Only 5 of 13 studies (38%) reported systematically performing eye dilation as part of their methodology, although an additional 5 of 13 (38%) utilized exclusively non-mydriatic fundus cameras designed for image acquisition without requirement for pupil dilation. Sporadic CSVD studies utilizing fundus photography identified arterial and venular fractal dimensions [22–24] or arteriolar and venular caliber [25,27,30] as associated with WMH, lacunar infarcts, or cerebral microbleeds. Four studies [27–30] reported

**Table 3. Summary of design, patient characteristics, methodology and results for studies of sporadic CSVD.**

| Study | Year | Design | CSVD Phenotype(s) | No. Participants | Imaging Modality | Device | Software | Pupil Dilation | Biomarkers Identified |
|---|---|---|---|---|---|---|---|---|---|
| Abdelhak et al. | 2020 | Case-Control | Lacunar Stroke VCID | 24 subjects 20 controls | OCT | Heidelberg Spectralis | Heidelberg Spectralis Version 6.2 | No | • Retinal Artery Mean Wall Thickness • Retinal Artery Wall to Lumen Ratio |
| Cheung et al. | 2010 | Cross-sectional | Lacunar Ischemic Stroke | 392 subjects | Fundus Photography | Fundus Camera (Not specified) | Semi-automated program (International Retinal Imaging Software [IRIS-Fractal]) | Yes | • Retinal Vessel fractal dimension |
| Doubal et al. | 2010 | Case-Control | Lacunar Ischemic Stroke | 86 subjects 80 controls | Fundus Photography | CRDGi Canon | Matlab | Yes | • Retinal Vessel fractal dimension |
| McGrory et al. | 2019 | Cross-sectional | Ischemic Stroke Lacunar Infarct WMH | 758 subjects | Fundus Photography | CRDGi Canon | VAMPIRE | No (non-mydriatic camera) | • Arteriolar fractal dimension • Venular fractal dimension |
| Hilal et al. | 2014 | Cross-sectional | Lacunar Infarcts WMH Cerebral Microbleeds | 261 subjects | Fundus Photography | Fundus Camera (Not specified) | Semi-automated program (Singapore I Vessel Assessment (v3.0)) | Yes | • Retinal Vessel fractal dimension • Retinal Vessel tortuosity |
| Ikram et al. | 2006 | Cohort | Lacunar Infarcts WMH | 490 subjects | Fundus Photography | Topcon Camera | Semi-automated program (Retinal Analysis, Optimate) | Yes | • Retinal Venular Dilation |
| Kim et al. | 2011 | Cohort | Lacunar Infarcts WMH | 4395 subjects | Fundus Photography | EOS D60 Canon Camera | N/A | No (non-mydriatic camera) | • RNFL Wedge-like Defect |
| Kwa et al. | 2002 | Cohort | Lacunar Infarct WMH | 108 subjects | Fundus Photography | Optimed | N/A | Yes | • Retinal microvascular abnormalities • Retinal arterial narrowing • Retinal arterial sclerosis • Retinal exudates |
| Lee et al. | 2019 | Cross-sectional | VCID | 1077 subjects 1547 controls | Fundus Photography | Fundus Camera (Not specified) | Only retinal arteriolar diameter calculated via semi-automated system | No (non-mydriatic camera) | • Retinopathy • Retinal microaneurysms • Retinal hemorrhages • AV nicking |
| Lee et al. | 2020 | Cross-sectional | AD Cognitive Impairment VCID | 60 subjects | OCT OCTA | Topcon DRI Triton | Native OCT software (IMAGEnet 6 V.1.14.8538) | No | • Retinal Capillary Density |
| Shu et al. | 2020 | Cross-sectional | Ischemic stroke or TIA | 263 subjects | Fundus Photography | KOWA nonmyd7 | N/A | No (non-mydriatic camera) | • Retinopathy score • Retinal microvascular abnormalities |
| Yatsuya et al. | 2010 | Cohort | Ischemic stroke | 10,496 subjects with 338 incident strokes | Fundus Photography | Canon CR-45UAF | "Computer-assisted approach" | No (non-mydriatic camera) | • Retinal Arteriolar Diameter • Retinal Venular Diameter • Retinal Focal Arteriolar Narrowing • AV nicking • Retinal microvascular abnormalities |

(*Continued*)

**Table 3.** (Continued)

| Study | Year | Design | CSVD Phenotype(s) | No. Participants | Imaging Modality | Device | Software | Pupil Dilation | Biomarkers Identified |
|---|---|---|---|---|---|---|---|---|---|
| Wang et al. | 2021 | Case-Control | Lacunar Infarcts WMH | 47 subjects 30 controls | OCTA | RTVue-XR OptoVue AVANTI | Native OCT software (AVANTI) | No | • Macular Superficial Capillary Plexus Vessel Density<br>• Radial Peripapillary Capillary Vessel Density |

Abbreviations: CSVD = Cerebral Small Vessel Disease, OCT = Optical Coherence Tomography, OCTA = Optical Coherence Tomography Angiography,

TIA = Transient Ischemic Attack, VCID = Vascular Contributions to Cognitive Impairment and Dementia, WMH = White Matter Hyperintensities.

associations between retinal markers of retinopathy (retinal hemorrhages, AV nicking, microvascular abnormalities) as associated with WMH and/or lacunar infarcts. A single large study [26] reported higher prevalence among CSVD patients (compared to healthy controls) of RNFL wedge-shaped defects on fundus photography, a semi-quantitative marker of focal nerve fiber damage [69]. Regarding OCT imaging, one study [51] reported no associations with retinal measurements, while another [52] reported increased arteriolar thickness, quantified as Mean Wall Thickness (MWT) or Wall-to-Lumen Ratio (WLR), among CSVD patients compared to controls. The latter report also identified an association between arteriolar WLR and WMH severity on MRI, as well as with select cerebrospinal fluid biomarkers. Sporadic CSVD studies collecting OCTA images [51,59] found lower retinal capillary density in the peripapillary network in patients with CSVD, which was also associated with WMH on MRI.

## Retinal biomarkers in CAA

We identified a total of three studies investigating the association between retinal biomarkers and CAA (Table 4). One study employed fundus photography, OCT and OCTA

**Table 4. Summary of design, patient characteristics, methodology and results for studies of CAA.**

| Study | Year | Design | CSVD Phenotype(s) | No. Participants | Imaging Modality | Device | Software | Pupil Dilation | Biomarkers Identified |
|---|---|---|---|---|---|---|---|---|---|
| Alber et al. | 2021 | Case-Control | Sporadic CAA Cerebral Microbleeds WMH Memory Performance | 12 subjects 12 controls | Fundus Photography | RTVue-XR OptoVue AVANTI | Native OCT software (AVANTI) | Yes | • Retinal hemorrhages |
| | | | | 12 subjects 12 controls | OCT | RTVue-XR OptoVue AVANTI | Native OCT software (AVANTI) | Yes | • None |
| | | | | 12 subjects 12 controls | OCTA | RTVue-XR OptoVue AVANTI | Native OCT software (AVANTI) | Yes | • None |
| Lee et al. | 2009 | Cross-Sectional | Sporadic CAA | 7 patients | Fundus Photography | N/A | N/A | Yes | • Retinal hemorrhages<br>• Retinal microaneurysms |
| | | | | 7 patients | Fluorescein Angiography | N/A | N/A | Yes | • Retinal hemorrhages<br>• Retinal microaneurysms. |
| van Etten et al. | 2020 | Case-Control | Hereditary (Dutch Mutation) CAA | 21 subjects 9 controls | Fundus Photography | Topcon TRC-50DX | N/A | Yes | • Retinal Arteriolar Narrowing |
| | | | | 21 subjects 9 controls | OCT | Heidelberg Spectralis | Heidelberg Eye Explorer v1.9.10.0 | Yes | • RNFL thickness |

Abbreviations: CAA = Cerebral Amyloid Angiopathy, CSVD = Cerebral Small Vessel Disease, OCT = Optical Coherence Tomography, OCTA = Optical Coherence Tomography Angiography, WMH = White Matter Hyperintensities.

concomitantly in a case-control design of 12 patients with possible or probable CAA (based on the validated Boston criteria) and 12 healthy controls [31]. Although this study identified no differences in retinal biomarkers between CAA cases and controls, retinal microbleeds were associated with episodic memory performance among CAA patients. Another study combined fundus photography with FA to examine a consecutive series of seven patients admitted with CAA-related intracerebral hemorrhage (as defined using the Boston criteria) [32]. Investigators found multiple dot and blot retinal hemorrhages on fundus photography and retinal microaneurysm in at least one eye for each CAA patient. The third study jointly employed fundus photography and OCT to conduct a case-control analysis of 21 carriers of the Dutch-mutation variant of Hereditary CAA (8 pre-symptomatic individuals without history of stroke or cognitive decline, and 13 symptomatic patients) and 9 healthy controls [33]. Retinal arteriolar narrowing was more common among mutation carriers (both symptomatic and asymptomatic) than controls. Peripapillary RNFL thickness was lower in symptomatic patients compared to controls, but not among pre-symptomatic individuals.

## Retinal biomarkers in fabry disease

We identified a total of 19 studies of retinal biomarkers in Fabry disease (Table 5). These studies included in total 558 affected individuals and 303 healthy controls. The median number of Fabry disease patients per study was 28 (range 8–57) and the median number of healthy

**Table 5. Summary of design, patient characteristics, methodology and results for studies of Fabry disease.**

| Study | Year | Design | CSVD Phenotype(s) | No. Participants | Imaging Modality | Device | Software | Pupil Dilation | Biomarkers Identified |
|---|---|---|---|---|---|---|---|---|---|
| Atiskova et al. | 2019 | Case-control | Fabry Disease | 27 subjects 27 controls | Fundus Photography | Heidelberg Spectralis | ImageJ | No | • Macular hyper-reflective foci • Retinal vessel tortuosity |
| | | | | 27 subjects 27 controls | OCT | Heidelberg Spectralis | ImageJ | Yes | • None |
| Bacherini et al. | 2021 | Case-control | Fabry Disease | 13 subjects 13 controls | OCTA | Nidek RS-3000 Advance 2 | Native OCT software (AngioScan) | Yes | • Superficial capillary plexus vessel density • Deep capillary plexus vessel density |
| Cakmak et al. | 2020 | Case-control | Fabry Disease | 25 subjects 37 controls | OCTA | RTVue-XR OptoVue Avanti | Native OCT software (AVANTI v 2018.0.0.18) | Yes | • Superficial capillary plexus vessel density • Deep capillary plexus vessel density • Foveal avascular zone area |
| Cennamo et al. | 2019 | Case-control | Fabry Disease | 54 subjects 70 controls | OCTA | RTVue-XR OptoVue Avanti | Native OCT software (AngioAnalytic) | No | • Superficial capillary plexus vessel density • Deep capillary plexus vessel density |
| Cennamo et al. | 2020 | Cross-sectional | Fabry Disease | 50 subjects | OCTA | RTVue-XR OptoVue Avanti | Native OCT software (ReVue XR v2017.1.0.151 & AngioAnalytic) | No | • None |
| Dogan et al. | 2020 | Case-control | Fabry Disease | 38 subjects 40 controls | OCTA | RTVue-XR OptoVue Avanti | Native OCT software (AVANTI v 2016.2.0) | Yes | • Central macular thickness • Deep capillary plexus vessel density • Choriocapillaris flow area. |

*(Continued)*

**Table 5.** (Continued)

| Study | Year | Design | CSVD Phenotype(s) | No. Participants | Imaging Modality | Device | Software | Pupil Dilation | Biomarkers Identified |
|---|---|---|---|---|---|---|---|---|---|
| Fledelius et al. | 2015 | Cross-sectional | Fabry Disease | 37 subjects | Fundus Photography | N/A | N/A | Yes | • CRAO<br>• Retinal arterial narrowing<br>• Retinal arterial tortuosity<br>• Retinal venous tortuosity |
| Lin et al. | 2021 | Case-control | Fabry Disease | 26 subjects 28 controls | OCT | Zeiss Cirrus HD 5000 | Native OCT software (Angioplex) | Yes | • Choroidal thickness |
| | | | | 26 subjects 28 controls | OCTA | Zeiss Cirrus HD 5000 | Native OCT software (Angioplex) | Yes | • Macular vessel length<br>• Superficial capillary plexus vessel density<br>• Superficial capillary plexus vessel length |
| Michaud | 2019 | Cross-sectional | Fabry Disease | 28 subjects | Fundus Photography | Canon Camera | N/A | Yes | • Retinal vessel tortuosity |
| | | | | 28 subjects | Visual Field | FDT Welch Allyn | N/A | Yes | • Unspecified VF defects |
| Minnella et al. | 2019 | Case-control | Fabry Disease | 20 subjects 17 controls | Fundus Photography | Topcon DRI Triton | Matlab | Yes | • Retinal vessel tortuosity |
| | | | | 20 subjects 17 controls | OCT | Topcon DRI Triton | Native OCT software | Yes | • None |
| | | | | 20 subjects 17 controls | OCTA | Topcon DRI Triton | Native OCT software | Yes | • Superficial capillary plexus vessel density<br>• Deep capillary plexus vessel density<br>• Perifoveal blood flow<br>• Foveal avascular zone area |
| | | | | 20 subjects 17 controls | ERG | N/A | N/A | Yes | • Decreased ERG response amplitude |
| Morier et al. | 2010 | Cross-sectional | Fabry Disease | 23 subjects | Fundus Photography | Kowa AD 5mp camera | N/A | Yes | • Retinal vessel tortuosity |
| | | | | 23 subjects | OCT | Zeiss Stratus | N/A | Yes | • None |
| San Román et al. | 2017 | Cross-sectional | Fabry Disease | 10 subjects | Fundus Photography | Zeiss Visucam Pro | Custom engineered software [40] | Yes | • Retinal vessel tortuosity<br>• Retinal venous tortuosity<br>• Retinal arterial tortuosity |
| | | | | 10 subjects | OCT | Zeiss Cirrus HD 5000 | N/A | Yes | • None |
| Sodi et al. | 2013 | Case-control | Fabry Disease | 35 subjects 35 controls | Fundus Photography | Zeiss TF 450 Plus Canon CF 60 UVI Topcon TRC-50VT | Custom engineered software | Yes | • Retinal vessel tortuosity |
| Sodi et al. | 2019 | Case-control | Fabry Disease | 11 subjects 11 controls | Fundus Photography | Zeiss TF 450 Plus | Custom engineered software [40] | Yes | • Retinal vessel tortuosity<br>• Retinal venous tortuosity<br>• Retinal arterial tortuosity |
| Sodi et al. | 2020 | Cross-sectional | Fabry Disease | 18 subjects | Fundus Photography | Imagine Eyes rtx1, i2k Align Retina software | Native rtx1 camera software | No | • Retinal Venous Tortuosity |

(*Continued*)

**Table 5.** (Continued)

| Study | Year | Design | CSVD Phenotype(s) | No. Participants | Imaging Modality | Device | Software | Pupil Dilation | Biomarkers Identified |
|---|---|---|---|---|---|---|---|---|---|
| Sodi et al. | 2021 | Case-control | Fabry Disease | 8 subjects 8 controls | Fundus Photography | N/A | Custom engineered software [40] | Yes | • Retinal arterial diameter |
| Wiest et al. | 2021 | Cross-sectional | Fabry Disease | 57 subjects | Fundus Photography | Optos | N/A | No | • Retinal vessel tortuosity |
| | | | | 57 subjects | OCTA | Zeiss PLEX Elite 9000 | Native OCT software (v2.0.1.47652; Macular Density v0.7.1) | No | • Retinal vessel tortuosity • Superficial capillary plexus vessel density |
| Orssaud et al. | 2003 | Cross-sectional | Fabry Disease | 32 subjects | Visual Field | Goldmann | N/A | Yes | • Enlarged blind spot |
| Pitz et al. | 2009 | Cross-sectional | Fabry Disease | 31 subjects | Visual Field | HVF 30–2 | N/A | No | • Heterogenous VF defects |

Abbreviations: CSVD = Cerebral Small Vessel Disease, ERG = Electroretinography, OCT = Optical Coherence Tomography, OCTA = Optical Coherence Tomography Angiography, WMH = White Matter Hyperintensities.

controls per study was 27 (range 8–70). Among included studies, 11 employed fundus photography [34–44], five used OCT [34,37–39,53], eight used OCTA [37,44,53,60–64], three leveraged VF assessment [36,67,68] and one presented results of ERG testing [37]. FA and VEP were the only imaging methodology not employed in published reports. Adequately detailed information on device and methodology used was provided by 17 of 19 studies (89%), and eye dilation was performed in 15 of 19 studies (79%). In studies using fundus photography, investigators repeatedly found associations between Fabry disease and several retinal vascular abnormalities, including retinal vessel tortuosity [34,39,40], retinal arteriolar narrowing [35], and decreased retinal arteriolar diameter [43]. There were no retinal biomarkers emerging as associated with Fabry disease diagnosis or severity in the five identified studies incorporating OCT imaging. Among eight studies conducting OCTA imaging, vessel density and foveal avascular zone area were most frequently reported as associated with Fabry disease diagnosis or severity. Six studies found decreased vessel density in the deep and/or superficial capillary plexus in Fabry patients [37,44,53,60,61,64]. One study reported vessel density as increased in the deep capillary plexus and decreased in the superficial capillary plexus [63]. Another study found no association between any retinal vessel density metrics and Fabry disease [62]. Regarding foveal avascular zone area, three papers [53,60,64] reported no difference between Fabry cases and controls and two papers [37,61] reported enlargement in affected individuals. Less frequently reported OCTA biomarkers found to be associated with Fabry disease were choriocapillaris flow area [64], perifoveal flow area [37], and macular vessel average length [53]. Studies incorporating VF assessment reported heterogenous abnormalities in Fabry disease patients, including multiple unspecified defects [36], blind spot enlargement [67,68], and scattered central scotomas [68]. One study of Fabry disease using ERG reported decreased in ERG mean amplitude among affected individuals [37].

## Retinal biomarkers in CADASIL

We identified 11 studies of retinal biomarkers in CADASIL, including 184 affected individuals and 142 healthy controls (Table 6). The median number of CADASIL patients per study was 30 (range 3–38) and the median number of controls per study was 16 (range 4–27). We identified five studies employing fundus photography [45–49], six employing OCT [45,48,54–57]

**Table 6. Summary of design, patient characteristics, methodology and results for studies of CADASIL.**

| Study | Year | Design | CSVD Phenotype(s) | No. Participants | Imaging Modality | Device | Software | Pupil Dilation | Biomarkers Identified |
|---|---|---|---|---|---|---|---|---|---|
| Alten et al. | 2014 | Case-control | CADASIL | 14 subjects 14 controls | Fundus Photography | Zeiss Visucam | ImageJ (semi-automated plugin) | No | • Arterio-venous nicking • Retinal venous dilation |
| | | | | 14 subjects 14 controls | OCT | Heidelberg Spectralis | Heidelberg Eye Explorer software | No | • RNFL thickness • Retinal vessel diameter |
| | | | | 14 subjects 14 controls | FA | Heidelberg Spectralis | N/A | No | • None |
| Cavallari et al. | 2011 | Cross-sectional | CADASIL WMH | 10 subjects 10 controls | Fundus Photography | N/A | ImageJ (with FracLac plugin) | No | • Retinal vessel fractal dimension |
| Cumurciuc et al. | 2004 | Cross-sectional | CADASIL | 18 subjects | Fundus Photography | N/A | N/A | No | • Heterogenous retinal abnormalities |
| | | | | 18 subjects | Visual Field | N/A | N/A | No | • No VF defects |
| | | | | 18 subjects | FA | N/A | N/A | No | • Heterogenous retinal findings |
| Fang et al. | 2017 | Case-control | CADASIL | 27 subjects 27 controls | OCT | Heidelberg Spectralis | Heidelberg Eye Explorer software | No | • Choroidal thickness • Retinal arterial diameter • Retinal venous diameter • Arterio-venous Wall thickness |
| Nelis et al. | 2018 | Case-control | CADASIL | 11 subjects 21 controls | OCT | Heidelberg Spectralis | Heidelberg Eye Explorer software | Yes | • None |
| | | | | 11 subjects 21 controls | OCTA | RTVue-XR OptoVue Avanti | ImageJ (v 1.51n) | Yes | • Deep capillary plexus vessel density |
| Parisi et al. | 2000 | Case-Control | CADASIL | 3 subjects 4 controls | ERG | BM 6000 Ganzfeld | Native BM 6000 software | Yes | • Delayed PERG responses |
| Parisi et al. | 2003 | Case-Control | CADASIL | 6 subjects 14 controls | ERG | BM 6000 Ganzfeld [66] | Native BM 6000 software [66] | Yes | • Delayed ERG, OP and PERG responses |
| Parisi et al. | 2007 | Case-control | CADASIL | 6 subjects 16 controls | OCT | Humphrey OCT3 | Native OCT software | Yes | • RNFL thickness |
| Pretegiani et al. | 2013 | Cross-sectional | CADASIL | 34 subjects | Fundus Photography | N/A | N/A | No | • Retinal arteriolar narrowing • Retinal venous dilation |
| | | | | 34 subjects | OCT | Zeiss Stratus 3000 OCT | N/A | No | • RNFL thickness |
| | | | | 34 subjects | Visual Field | Automated Perimetry or Goldmann Perimetry | N/A | No | • Heterogenous visual field defects |
| | | | | 34 subjects | FA | N/A | N/A | No | • Heterogenous retinal findings |
| | | | | 34 subjects | VEP | N/A | N/A | No | • Heterogeneous VEP abnormalities |
| Roine et al. | 2006 | Cross-sectional | CADASIL | 38 subjects 16 controls | Fundus Photography | N/A | Olympus DP-SOFT v 3.2 | Yes | • Retinal arteriolar narrowing • Arterio-venous ratios • Arterio-venous nicking |

*(Continued)*

**Table 6.** (Continued)

| Study | Year | Design | CSVD Phenotype(s) | No. Participants | Imaging Modality | Device | Software | Pupil Dilation | Biomarkers Identified |
|-------|------|--------|-------------------|------------------|------------------|--------|----------|----------------|-----------------------|
| Rufa et al. | 2011 | Case-control | CADASIL | 17 subjects 20 controls | OCT | Zeiss Stratus 3000 OCT | Peripapillary fast RNFL program (v 3.0) | Yes | • RNFL thickness |
| | | | | 17 subjects 20 controls | VEP | N/A | N/A | Yes | • Heterogeneous VEP abnormalities |

Abbreviations: CADASIL = Cerebral Autosomal Dominant Arteriopathy with Subcortical Infarcts and Leukoencephalopathy, CSVD = Cerebral Small Vessel Disease, ERG = Electroretinography, FA = Fluorescein Angiography, OCT = Optical Coherence Tomography, OCTA = Optical Coherence Tomography Angiography, OP = Oscillatory Potentials, PERG = Pattern Electroretinography, RNFL = Retinal Nerve Fiber Layer, VEP = Visual Evoked Potentials, WMH = White Matter Hyperintensities.

and one OCTA [55], two studies presenting VF assessment results [47,48], three including FA results [45,47,48], two conducting ERG [65,66], and two including VEP data [48,57]. We determined that 6 of 11 studies (55%) provided detailed information on both the device utilized and methodology. Pupil dilation was performed and adequately reported from a methodological standpoint in 6 of 11 studies (55%). Among fundus photography studies, microvascular abnormalities were found to be associated with CADASIL diagnosis, including, specifically, arteriolar narrowing [48,49] and AV nicking [45,49]. One study reported lower retinal vessel fractal dimensions in CADASIL cases compared to controls [46]. OCT imaging identified decreased RNFL thickness as associated with CADASIL in 3 of 6 studies, either in all quadrants [48,56] or specifically in the temporal quadrant [57]. A single OCTA study identified decreased vessel density in the deep retinal plexus in CADASIL patients compared to healthy controls [55]. In two cross-sectional studies performing VF assessment in CADASIL patients there were no specific abnormalities found to be consistently present in affected individuals, though a number of isolated heterogenous abnormalities were reported [47,48]. Similarly, the results of three studies employing FA were notable for isolated findings (RPE changes, scattered drusen) in a handful of affected individuals [45,47,48]. Two case-control studies employing ERG identified delayed ERG, oscillatory potential (OP) and pattern ERG (PERG) responses [65,66]. Both studies employing VEP reported asymmetric P100 latency and bilateral increase in P100 delay, but these findings were present in less than half of affected individuals [48,57].

## Retinal biomarkers in MELAS

We found two studies investigating retinal biomarkers in MELAS (Table 7). An older study presents fundus photography and ERG data from 26 individuals with genetically confirmed MELAS diagnosis [50]. The investigators identified paramacular RPE atrophy in 10 of 26 patients (38%), and found decreased ERG response amplitude, increased latency, or both in 7 of 8 patients who underwent electrodiagnostic evaluation (88%) A more recent study performed OCT imaging on 10 affected individuals and 5 healthy controls [58]. The investigators found lower GCL thickness among MELAS patients compared to controls (after adjusting for prior episodes of transient homonymous hemianopia potentially accounting for direct retinal disease involvement). Lower GCL thickness was also associated with longer disease duration among affected individuals.

## Discussion

In this systematic review we identified 48 studies investigating associations between retinal biomarkers and different forms of CSVD, including a total of 21423 participants (9147 CSVD

**Table 7. Summary of design, patient characteristics, methodology and results for studies of MELAS.**

| Study | Year | Design | CSVD Phenotype(s) | No. Participants | Imaging Modality | Device | Software | Pupil Dilation | Biomarkers Identified |
|---|---|---|---|---|---|---|---|---|---|
| Latvala et al. | 2002 | Cross-sectional | MELAS | 26 subjects | Fundus Photography | Canon FC-60Z connected to Kodak digital camera system | N/A | Yes | • RPE atrophy |
| | | | | 8 subjects | ERG | Nicolet Viking II | N/A | Yes | • Decreased ERG amplitude • Increased ERG latency |
| Shinkai et al. | 2021 | Cross-sectional | MELAS | 10 subjects 5 controls | OCT | Nidek RS-3000 Advance | Native OCT software (v 1.5.5.0) | No | • GCL thickness |

Abbreviations: CSVD = Cerebral Small Vessel Disease, ERG = Electroretinography, MELAS = Mitochondrial Encephalomyopathy, Lactic Acidosis, and Stroke-like episodes syndrome, OCT = Optical Coherence Tomography.

cases and 12276 healthy controls). Overall, our review identified multiple reported associations between retinal biomarkers and CSVD-related clinical outcomes and neuroimaging metrics. From a purely theoretical standpoint, retinal biomarkers could therefore replace (or at least complement) neuroimaging in initial diagnosis and longitudinal follow-up of CSVD, owing to: 1) lower costs associated with acquisition and operation of retinal evaluation scanners vs. MRI scanners; 2) availability of multiple options (fundus photography, OCT, OCTA) for retinal evaluation using portable devices; 3) relative availability of personnel trained to perform retinal vs. neuroimaging evaluations, and interpret study results [8]. These advantages would also make screening of asymptomatic at risk individuals potentially feasible, in a way that MRI-based neuroimaging is currently not capable of. However, published evidence falls short of clearly quantifying the diagnostic performance sensitivity of these biomarkers; thus, we could not definitively assess their relevance and yield regarding future research studies and clinical practice. Therefore, our systematic review highlights the need for larger, more adequately powered and specifically designed studies in order to address these open questions.

We found substantial heterogeneity in sample size across included studies, ranging from small cross-sectional surveys including a handful of cases to large cohort studies with thousands of participants. Larger studies utilizing a cohort format would generally be expected to provide more robust information on the association between retinal biomarkers and CSVD. However, the large cohort studies included in our review exclusively utilized fundus photography for retinal evaluation, thus being unable to access insights provided by more recent studies employing OCT/OCTA to study neurodegenerative and neurovascular disorders [8]. Indeed, most participating studies employed a single retinal evaluation modality, with only a handful combining two or more modalities. Therefore, our findings point to the need for large, adequately powered studies of retinal biomarkers that utilize validated standardized methodologies for capture of CSVD-related clinical outcomes and neuroimaging markers [1]. Additional desirable features for planned future studies of retinal biomarkers in CSVD include the use of multiple retinal evaluation technologies and longitudinal evaluation of biomarkers (both retinal and neuroimaging) over time, to be correlated with CSVD clinical progression in the form of subsequent stroke risk and cognitive decline. Eventually, these longitudinal studies incorporating parallel, repeat retinal and brain imaging over time would also be instrumental in clarify the biological relationships existing between neuronal and microvascular changes occurring in different anatomical locations [8].

Our findings also emphasize the importance of adopting more standardized approaches to study design (with specific emphasis on longitudinal retinal evaluation and estimation of

adequate sample sizes on the basis of robust power calculations) and execution (emphasizing careful and detailed reporting of hardware, software, protocols, and data acquisition parameters employed). Of note, detailed recommendations for design, execution, and results' reporting are currently available only for OCT/OCTA studied (in the form of the APOSTEL 2.0 recommendations) [17]. Future CSVD studies employing other technologies would benefit from consensus-driven formulation of similar guidelines, which would in turn enhance scientific rigor and reproducibility of reported findings.

The vast majority of participants in this systematic review were enrolled in studies investigating sporadic CSVD. In all included reports, a diagnosis was made via a combination of clinical history (usually CSVD-related lacunar stroke) and neuroimaging (usually lacunes or white matter disease). However, none of the included studies conducted subtyping of sporadic CSVD to determine the relative prevalence of its two most common subtypes, CAA and HTNA. While we did identify dedicated studies of retinal biomarkers in CAA, HTNA has so far not yet been evaluated in-depth. In addition, variations in CSVD diagnostic criteria resulted in heterogeneity in clinical severity, ranging from asymptomatic, to minor stroke, to advanced cognitive decline or severe stroke. Finally, the overwhelming majority of studies focused only on WMH and lacunar infarcts as neuroimaging CSVD markers, while neglecting all others [7]. Taken together, published evidence supports an association between retinal microvasculature abnormalities and sporadic CSVD, whether quantified as discrete findings (microvascular abnormalities, defined as presence of hemorrhages, arteriolar narrowing, venular dilation, or AV nicking), vessel diameter, or fractal dimension. As previously mentioned, published studies have yet to provide reliable estimates for the diagnostic performance of retinal biomarkers in CSVD diagnosis or staging, a key prerequisite for more widespread application to research endeavors and introduction in clinical practice.

Despite the relevance of CAA as a major contributor to stroke incidence and cognitive decline [70,71], we found only three studies investigating retinal biomarkers in CAA that included 61 participants in total. While accounting for limitations due to very small sample size, these studies identified retinal hemorrhages as potentially sensitive markers of CAA and correlated them with hemorrhagic CNS disease burden. However, similar findings were identified upon reviewing studies focusing on sporadic CSVD at large, as mentioned above. It remains to be determined whether retinal hemorrhages represent specific retinal biomarkers for CAA (as opposed to HTNA) that were included in studies of CSVD at large due to incomplete clinical characterization. It is alternatively possible that retinal hemorrhages represent shared retinal biomarkers in all forms of CSVD, regardless of subtype. No OCT or OCTA derived retinal biomarker associated with CAA has emerged to date, although limited sample size and substantial differences in methodology and approaches are likely responsible. Indeed, larger, more adequately powered studies of CAA including at minimum fundus photography, OCT and OCTA are warranted based on findings from this systematic review.

Despite being an uncommon diagnosis in clinical practice, reports investigating retinal biomarkers in Fabry disease accounted for 40% of studies included in this systematic review, and included 861 participants in total. Investigators also reported on a wide array of retinal evaluation modalities for this CSVD subtype, with only fluorescein angiography and VEP lacking dedicated studies. Overall, our review findings indicate that retinal microvascular abnormalities are frequently identified in Fabry disease as either discrete abnormalities or decreased vessel density and may therefore represent sensitive biomarkers. However, similar findings were reported in studies of sporadic CSVD (as reported above) and may therefore not be specific to Fabry disease. Additional studies are warranted to expand upon these observations and categorize retinal vascular biomarkers in a systematic fashion, ideally combining different methodologies in each study to increase likelihood of identifying patterns specific to this condition.

We found a substantial number of studies (23% of total) investigating retinal biomarkers in CADASIL. This CSVD disorder was also the only condition with published reports for all retinal evaluation modalities considered in this systematic review. Taken together, available evidence indicates that retinal microvascular changes and decreased RNFL thickness may represent sensitive markers for CADASIL. As previously discussed for Fabry disease, these or very similar findings have also been reported in sporadic CSVD, raising concerns about their diagnostic specificity. Electrodiagnostic studies (ERG and VEP) also uncovered a variety of abnormal findings in CADASIL patients, though not consistently and only in a subset of affected individuals. Larger studies are warranted to clarify the diagnostic performance of retinal vascular measures in CADASIL and to systematically assess the relevance of previously identified electrodiagnostic abnormalities.

We included MELAS as a CSVD disorder in our systematic review on the basis of prior reports implicating small vessel vasculopathy in the pathogenesis of stroke associated with this condition. We identified only two studies of retinal biomarkers in MELAS that did not find definitive associations. Therefore, there is currently no evidence as to whether MELAS-related small vessel vasculopathy can be identified in the retina. Currently published findings (albeit limited in terms of small sample size) support the hypothesis of retinal involvement in MELAS, thus warranting additional, larger studies of its impact on neuronal and vascular biomarkers.

## Conclusions

In this systematic review we identified associations between several retinal biomarkers and CSVD-related clinical outcomes (stroke and cognitive impairment) and neuroimaging findings (chiefly white matter disease, lacunes, and cerebral microbleeds). Retinal microvascular abnormalities identified via either fundus photography, OCT or OCTA have so far generated the largest amount of published evidence for association with CSVD. However, definitive evidence on the performance of retinal biomarkers in diagnosing CSVD and following its progression over time is currently lacking. The majority of published studies also suffered from several methodological limitations, chiefly small sample sizes and inadequate reporting of key factors in study design, protocols for data capture, and analytical methods. Larger, adequately powered studies employing standardized methodologies for both retinal evaluation and CSVD characterization (ideally incorporating repeated measurements over time) are therefore required to definitively establish the potential impact of these technologies in future research efforts and clinical practice.

## Supporting information

**S1 Checklist. Retinal imaging in CSVD review—PRISMA checklist.**
(PDF)

**S1 File. Search strategy.** Terms, syntax, and databases used to perform systematic review of scientific literature.
(DOCX)

## Acknowledgments

All authors had substantial contributions in the conception, design, analysis, or interpretation of the work. All authors contributed to the draft and revision of the content, and final approval version to be published.

## Author Contributions

**Conceptualization:** Elena Biffi, Zachary Turple, Jessica Chung, Alessandro Biffi.

**Data curation:** Elena Biffi, Zachary Turple, Jessica Chung, Alessandro Biffi.

**Formal analysis:** Elena Biffi, Zachary Turple, Jessica Chung, Alessandro Biffi.

**Funding acquisition:** Elena Biffi, Alessandro Biffi.

**Investigation:** Elena Biffi, Zachary Turple, Jessica Chung, Alessandro Biffi.

**Methodology:** Elena Biffi, Zachary Turple, Jessica Chung, Alessandro Biffi.

**Project administration:** Elena Biffi.

**Resources:** Elena Biffi.

**Software:** Elena Biffi.

**Supervision:** Elena Biffi, Alessandro Biffi.

**Validation:** Elena Biffi, Alessandro Biffi.

**Visualization:** Elena Biffi, Alessandro Biffi.

**Writing – original draft:** Elena Biffi, Zachary Turple, Jessica Chung, Alessandro Biffi.

**Writing – review & editing:** Elena Biffi, Zachary Turple, Jessica Chung, Alessandro Biffi.

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
