## [Decision Letter · Decision Letter 0]

16 Feb 2022

PONE-D-22-00086Retinal Imaging Biomarkers of Cerebral Small Vessel Disease:a Systematic ReviewPLOS ONE

Dear Dr. Biffi,

Thank you for submitting your manuscript to PLOS ONE. After careful consideration, we feel that it has merit but does not fully meet PLOS ONE’s publication criteria as it currently stands. Therefore, we invite you to submit a revised version of the manuscript that addresses the points raised during the review process.

The authors need to pay attention to the interpretation of their findings.

We look forward to receiving your revised manuscript.

Kind regards,

Rayaz A Malik, MBChB, PhD

Academic Editor

PLOS ONE

Journal Requirements:

"This study was supported by NIH T35EY007149, and by grants from the National Academy of Medicine and the American Optometric Association. The funding entities played no role in the study design, data collection and analysis, decision to publish, or preparation of the manuscript."

We note that you have provided funding information. However, funding information should not appear in the Funding section or other areas of your manuscript. We will only publish funding information present in the Funding Statement section of the online submission form. 

"This study was supported by NIH T35EY007149, and by grants from the National Academy of Medicine and the American Optometric Association. The funding entities played no role in the study design, data collection and analysis, decision to publish, or preparation of the manuscript."

Additional Editor Comments:

Whilst your work has merit, reviewer 2 in particular has identified several areas which need to be addressed.

Reviewers' comments:

Reviewer's Responses to Questions

**Comments to the Author**

1. Is the manuscript technically sound, and do the data support the conclusions?

Reviewer #1: Yes

Reviewer #2: Yes

2. Has the statistical analysis been performed appropriately and rigorously? 

Reviewer #1: N/A

Reviewer #2: I Don't Know

3. Have the authors made all data underlying the findings in their manuscript fully available?

Reviewer #1: Yes

Reviewer #2: No

4. Is the manuscript presented in an intelligible fashion and written in standard English?

Reviewer #1: Yes

Reviewer #2: Yes

5. Review Comments to the Author

Reviewer #1: Elena Biffi and coworkers performed a systematic review on using retinal imaging biomarkers (obtained with different image modalities) for correlation with cerebral small vessel disease. The authors evaluated non-invasive retinal imaging in the context of “eye window to the brain”. The authors discuss in their report 48 studies that passed their quality controls and inclusion criteria. Several associations were found between a single retinal imaging modality and cerebral vessel disease. Although promising, the authors suggest that much more work is needed before the technique can be implemented in the clinical routine. This reviewer agrees that retinal imaging can provide relevant assessments, but clinical validity and clinical utility need to be demonstrated first.

The study has its merits because it used a systematic approach, and it follows good practices for executing a systematic review. The text is well written and comprehensively describes the findings. However, the text may benefit from a more detailed assessment of the findings and interpretations. This will make the text more informative for the reader. The text, in many instances, only sums the observations. Considering the initial work, the authors should be allowed to revise the manuscript.

Some specific suggestions:

Abstract

The abstract indicates that there are still many hurdles, but the main text mentions that “promising” and “reliable” biomarkers are identified at several locations. This is not in alignment, and I suggest revising the main text, making it more neutral, and describing more objectively what the exact status is rather than using generic terms such as “promising” and “reliable”

Introduction

• I find this statement confusing this “The vast majority of CSVD cases are sporadic in nature, presenting without a clear familial inheritance pattern as the two primary subtypes of Cerebral Amyloid Angiopathy (CAA) and Hypertensive Arteriopathy (HTNA).” Perhaps the authors can give a graphical overview of the diseases, including reference to prevalence; incidence,…

• What is meant with “reliable”. Please specify the criteria. The specific application domain will probably require different reliability criteria. Can the authors comment on this and include a paragraph?

• Where in the disease development/follow-up could retinal biomarkers be used? What is the added-value apart from being “non-invasive”. The gold standard is brain imaging? Are there any shortcomings? What other non-invasive biomarkers are under scrutiny? How does retina compare to them? Can the authors spend attention to this or comment in their manuscript?

• The statement :”Our primary goal is to identify retinal imaging biomarkers which reliably differentiate patients diagnosed with CSVD (whether sporadic or familial) from healthy controls.” is somewhat confusing for this reviewer. Is this indeed the primary objective? Differentiating between CSVD patients and healthy controls is perhaps not that difficult, once patients are identified. Is this the main reason why you would use retinal imaging metrics?

Materials and Methods

Why do the authoers others refer specifically to APOSTEL recommendations? Please clarify? Why is this needed? Please clarify this in the text. Are there any quality assessments to be done for the other technologies? Does it exist? Would it be useful. Perhaps the authors can comment this in the discussion section.

Results

• Wat do the authors mean with “trends should be further clarified. There were no biomarkers displaying consistent associations across all CSVD disorders of interest.” Is this needed or expected?

• Interestingly, the authors report on scores for study quality assessment. This is useful, but apart from the scores, no further explanation is given. The authors should report on what this means? Are the studies reliable or not? Are all items in the score equally important?

• The results section reports on parameters such as tortuosity and fractal parameters, but no further explanation is given about what is means and the context.

• Throughout the text, this reviewer noticed several times that “trend” or “correlation” is given, but no information is given about the directionality (positive/negative), strength of association,…. What does “trend” mean? This type of information is very relevant for interpretation and should be added throughout the text.

• This reviewer is aware of cross-sectional studies and prospective studies focusing on retinal vessel metrics and stroke. (for example the extensive studies involving Prof. T.Y. Wong). It is important to make differentiation between these different study designs because they will give different evidence for using retinal biomarkers. The authors will need to clarify this.

• The review is systematic and Tables 2-6 are comprehensive. Nevertheless, this reviewer thinks the manuscript will benefit from a visual representation giving the most essential information. The authors should interpret and digest the manuscripts and report on the trends in a figure to help the reader to keep the overview; comparable to a graphical abstract. Can the authors please consider this.

• It seems there is a large diversity in types of publications, from small case-control to large studies. The authors make no difference between them. However, the weight of evidence large studies give is much higher. This should be clearly discussed.

Discussion

• The conclusion starts by saying “Overall, our review identified several promising retinal biomarkers of potential value in diagnosis and monitoring of CSVD. However, published evidence falls short of clearly quantifying the sensitivity and specificity of these biomarkers; thus, we could not definitively assess their relevance and yield regarding future research studies and clinical practice.” Why do the authors consider it then “promising”? The criteria are not clearly given. Why are the authors enthusiastic?

• It is clear that type of instruments, imaging modalities, image algorithms and type of metrics used have an import consequence on the outcome and future generalizability of the results. The authors should comment on this. Are there any studies they looked into that confirmed results or are all studies stand-alone?

• The authors could discuss more on the mechanism of retinal changes in relation to brain disease? Are retinal changes occurring in parallel of brain diseases as result of a common disease mechanism, making retinal markers a potential proxy or are retinal changes a consequence?

• Can the authors comment on the envisioned use of future retinal markers: diagnostic marker, prognostic marker, stratification,… what could be the clinical use and what would be needed to reach this? What are the most promising avenues? Clearly, retinal biomarkers have their pro- and cons.

• This reviewer is of the opinion that the conclusion should be more specific. It now reads very generic: “In this systematic review we identified several promising retinal imaging biomarkers of CSVD. Retinal microvascular abnormalities identified via either fundus photography, OCT or OCTA have so far generated the largest amount of published evidence for association with CSVD. However, larger studies employing standardized methodologies for both retinal imaging and CSVD characterization are required to definitively establish the potential impact of these technologies in future research efforts and clinical practice.”

Reviewer #2: Biffi et al have written an important systematic review “Retinal imaging biomarkers of cerebral small vessel disease”. Addition of visual fields, electroretinography, and visual evoked potentials along with OCTA, OCT, fundus photograph have added more value to the review, however, they don’t seem to be imaging modalities. Therefore, retinal biomarkers….seem to be a better title.

I have few minor comments.

Please expand the abbreviations in the abstract. CAA, or HTNA.

Can you please clearly define CAA/HTNA/CADASIL/MELAS for general readers in the introduction.

Adding the different software used to measure retinal metrics and/or criteria for classification of CSVD in the table would add more value to the review

6. PLOS authors have the option to publish the peer review history of their article (what does this mean?). If published, this will include your full peer review and any attached files.

Reviewer #1: No

Reviewer #2: No

---

## [Author Response · Author response to Decision Letter 0]

20 Mar 2022

Retinal Biomarkers of Cerebral Small Vessel Disease: a Systematic Review

Elena ZB et al.

We would like to thank the Editors at PLOS ONE for giving us the opportunity to submit a revised version of this manuscript, and to the reviewers for their thoughtful criticism of our work. With their guidance, we have modified our original submission to clarify the relevance significance of our findings. Please find below itemized replies to the reviewer’s suggestions and comments.

Editorial and Journal Requirements:

We have included all required items in our resubmission.

No changes are required to our financial disclosures.

Journal Requirements:

"This study was supported by NIH T35EY007149, and by grants from the National Academy of Medicine and the American Optometric Association. The funding entities played no role in the study design, data collection and analysis, decision to publish, or preparation of the manuscript." We note that you have provided funding information. However, funding information should not appear in the Funding section or other areas of your manuscript. We will only publish funding information present in the Funding Statement section of the online submission form. Please remove any funding-related text from the manuscript and let us know how you would like to update your Funding Statement.

Currently, your Funding Statement reads as follows: 

"This study was supported by NIH T35EY007149, and by grants from the National Academy of Medicine and the American Optometric Association. The funding entities played no role in the study design, data collection and analysis, decision to publish, or preparation of the manuscript."

We have modified our submission to comply with journal requirements. As previously mentioned, no changes are required to our financial disclosures.

Additional Editor Comments:

Whilst your work has merit, reviewer 2 in particular has identified several areas which need to be addressed.

Please find below itemized responses to reviewers’ comments.

Reviewer #1:

Elena Biffi and coworkers performed a systematic review on using retinal imaging biomarkers (obtained with different image modalities) for correlation with cerebral small vessel disease. The authors evaluated non-invasive retinal imaging in the context of “eye window to the brain”. The authors discuss in their report 48 studies that passed their quality controls and inclusion criteria. Several associations were found between a single retinal imaging modality and cerebral vessel disease. Although promising, the authors suggest that much more work is needed before the technique can be implemented in the clinical routine. This reviewer agrees that retinal imaging can provide relevant assessments, but clinical validity and clinical utility need to be demonstrated first.

The study has its merits because it used a systematic approach, and it follows good practices for executing a systematic review. The text is well written and comprehensively describes the findings. However, the text may benefit from a more detailed assessment of the findings and interpretations. This will make the text more informative for the reader. The text, in many instances, only sums the observations. Considering the initial work, the authors should be allowed to revise the manuscript.

Some specific suggestions:

We thank Reviewer 1 for their kind words. We have modified the revised version of our manuscript to provide additional context on our findings and in-depth interpretation of their relevance to the fields of CSVD care and research at large.

Please find below itemized responses to your queries and suggestions.

Abstract

The abstract indicates that there are still many hurdles, but the main text mentions that “promising” and “reliable” biomarkers are identified at several locations. This is not in alignment, and I suggest revising the main text, making it more neutral, and describing more objectively what the exact status is rather than using generic terms such as “promising” and “reliable”.

We have extensively modified the main text of the manuscript to convey that, while multiple associations have been reported between retinal biomarkers and CSVD a number of limitations in currently available evidence prevents their widespread adoption in clinical and research efforts.

Introduction

I find this statement confusing this “The vast majority of CSVD cases are sporadic in nature, presenting without a clear familial inheritance pattern as the two primary subtypes of Cerebral Amyloid Angiopathy (CAA) and Hypertensive Arteriopathy (HTNA).” Perhaps the authors can give a graphical overview of the diseases, including reference to prevalence; incidence,…

We have modified the Introduction section to more clearly illustrated the epidemiological landscape of CSVD, including the relative contribution of sporadic (i.e. non monogenic) and familial (monogenic) forms, and the relative prevalence of different etiological subtypes (Page 4, line 10 to Page 6, line 8). We also included a brief summary of different forms of CSVD in Table 1 in the revised manuscript.

What is meant with “reliable”. Please specify the criteria. The specific application domain will probably require different reliability criteria. Can the authors comment on this and include a paragraph?

We modified the Introduction section of the revised manuscript (Page 7, line 20 to Page 8, line 2) to clarify that our primary aim is to review studies identifying associations between CSVD biomarkers and disease status (case vs. control) and CSVD neuroimaging markers. We pre-specified a secondary goal of identifying studied providing detailed information on diagnostic performance of retinal biomarkers for CSVD disorders (of which none were identified.

Where in the disease development/follow-up could retinal biomarkers be used? What is the added-value apart from being “non-invasive”. The gold standard is brain imaging? Are there any shortcomings? What other non-invasive biomarkers are under scrutiny? How does retina compare to them? Can the authors spend attention to this or comment in their manuscript?

We have modified the Introduction section of the revised manuscript (Page 6, line 15 to Page 7, line 8) to: 1) clarify that neuroimaging currently represents the diagnostic gold standard; and 2) illustrate some of the potential applications, benefits, and drawbacks of retinal imaging in CSVD screening, diagnosis, and longitudinal monitoring of CSVD.

The statement :”Our primary goal is to identify retinal imaging biomarkers which reliably differentiate patients diagnosed with CSVD (whether sporadic or familial) from healthy controls.” is somewhat confusing for this reviewer. Is this indeed the primary objective? Differentiating between CSVD patients and healthy controls is perhaps not that difficult, once patients are identified. Is this the main reason why you would use retinal imaging metrics?

As mentioned above, we modified the Introduction section of the revised manuscript (Page 7, line 20 to Page 8, line 2) to clarify the scope of our review and the potential applications of retinal biomarkers in CSVD.

Materials and Methods

Why do the authors others refer specifically to APOSTEL recommendations? Please clarify? Why is this needed? Please clarify this in the text. 

The APOSTEL recommendations are the only guideline available to guide design, execution, and reporting of OCT biomarkers studies in clinical neuroscience. We have modified the Methods section of the revised manuscript (Page 11, lines 15 to 19) to present the relevance of the APOSTEL recommendations more clearly, and detail our rationale for referring to them in our assessment of study quality.

Are there any quality assessments to be done for the other technologies? Does it exist? Would it be useful. Perhaps the authors can comment this in the discussion section.

In response to this important comment from reviewer 1 we have modified the Methods (Page 11, lines 21 to 23) and Discussion (Page 35, line 16 to Page 36, line 1) of the revised manuscript to clarify that no further guidelines for retinal biomarker studies in neuroscience are currently available, representing a crucial need in the field.

Results

What do the authors mean with “trends should be further clarified. There were no biomarkers displaying consistent associations across all CSVD disorders of interest.” Is this needed or expected?

We modified the Results section (Page 15, line 23 to Page 16, line 4) of the revised manuscript to clarify that we sought to compare retinal biomarker findings across different forms of CSVD to gage whether consistent association patterns emerged - potentially indicating shared pathophysiological mechanisms. As we more clearly discuss in this revised section, lack of overlap in reported associations across different CSVD conditions is far more likely to reflect limitation of existing evidence (especially application of different technologies and methodologies) rather than true underlying biological heterogeneity.

Interestingly, the authors report on scores for study quality assessment. This is useful, but apart from the scores, no further explanation is given. The authors should report on what this means? Are the studies reliable or not? Are all items in the score equally important?

We modified the Results (Page 16, lines 14 to 19) and Discussion (Page 35, lines 16 to 20) sections of the revised manuscript to more clearly present the relevance of our quality assessment results, as well as identify areas of improvement for future studies.

The results section reports on parameters such as tortuosity and fractal parameters, but no further explanation is given about what is means and the context.

We modified the Results (Page 15, lines 16 to 20) to include additional information on the relevance of this parameters and provide an additional reference.

Throughout the text, this reviewer noticed several times that “trend” or “correlation” is given, but no information is given about the directionality (positive/negative), strength of association,…. What does “trend” mean? This type of information is very relevant for interpretation and should be added throughout the text.

We have extensively modified the Results section of the revised manuscript to identify associations between retinal biomarkers and outcomes of interest based on the directionality, magnitude, and significance of the association (where applicable).

This reviewer is aware of cross-sectional studies and prospective studies focusing on retinal vessel metrics and stroke. (for example the extensive studies involving Prof. T.Y. Wong). It is important to make differentiation between these different study designs because they will give different evidence for using retinal biomarkers. The authors will need to clarify this.

Of note, several cross-sectional and cohort studies investigating retinal imaging metrics were excluded from our review because they did not specifically focus on CSVD-related stroke and cognitive decline subtypes. We have modified the Methods section (Page 10, lines 1 to 6) to clarify this aspect of our approach.

The review is systematic and Tables 2-6 are comprehensive. Nevertheless, this reviewer thinks the manuscript will benefit from a visual representation giving the most essential information. The authors should interpret and digest the manuscripts and report on the trends in a figure to help the reader to keep the overview; comparable to a graphical abstract. Can the authors please consider this.

We have modified Figures 2 and 3 in the revised manuscript to provide additional information on overall findings uncovered in our review.

It seems there is a large diversity in types of publications, from small case-control to large studies. The authors make no difference between them. However, the weight of evidence large studies give is much higher. This should be clearly discussed.

We have modified the Discussion section (Page 34, line 19 to Page 35, line 4) to discuss in greater depth the relative merits and weaknesses of larger studies among those included in our review.

Discussion

The conclusion starts by saying “Overall, our review identified several promising retinal biomarkers of potential value in diagnosis and monitoring of CSVD. However, published evidence falls short of clearly quantifying the sensitivity and specificity of these biomarkers; thus, we could not definitively assess their relevance and yield regarding future research studies and clinical practice.” Why do the authors consider it then “promising”? The criteria are not clearly given. Why are the authors enthusiastic?

We have modified the Discussion section of the revised manuscript (Page 34, lines 4 to 19) to clarify that retinal biomarkers offer many appealing theoretical advantages compared to current neuroimaging-based evaluations of CSVD (easier to perform and repeat, cheaper, lower expertise level required). However, while currently available evidence indicates association with CSVD clinical and neuroimaging parameters of interest, we did not identify sufficient information to warrant their immediate incorporation in clinical practice. Further investigative efforts aimed at addressing the knowledge gaps we identified are therefore needed.

It is clear that type of instruments, imaging modalities, image algorithms and type of metrics used have an import consequence on the outcome and future generalizability of the results. The authors should comment on this. Are there any studies they looked into that confirmed results or are all studies stand-alone?

We have modified the Discussion section of the revised manuscript (Page 35, line 20 to Page 36, line 1) to more clearly convey that the vast majority of studies pose challenges in direct comparison because of difference in the parameters listed by Reviewer 1 (technology used, study format, imaging protocols, study populations, markers of interest) with limited opportunity for consistent replication of findings in the field.

The authors could discuss more on the mechanism of retinal changes in relation to brain disease? Are retinal changes occurring in parallel of brain diseases as result of a common disease mechanism, making retinal markers a potential proxy or are retinal changes a consequence?

Reviewer 1 does highlight a topic of great interest, which would however require a more in-depth evaluation of evidence across multiple neurological disorders, and is therefore somewhat outside the scope of our review. We have modified the Discussion section of the revised manuscript (Page 35, lines 12 to 15) to introduce this topic and refer to our recent review focusing more on the biological aspects of the relationship between retinal and brain pathology (Prog Retin Eye Res. 2021 Jul;83:100938.).

Can the authors comment on the envisioned use of future retinal markers: diagnostic marker, prognostic marker, stratification,… what could be the clinical use and what would be needed to reach this? What are the most promising avenues? Clearly, retinal biomarkers have their pro- and cons.

We have modified the Discussion section of the revised manuscript (Page 34, lines 4 to 19) to more clearly present the potential application of retinal biomarkers in CSVD, as well as present current and future potential benefits and drawbacks.

This reviewer is of the opinion that the conclusion should be more specific. It now reads very generic: “In this systematic review we identified several promising retinal imaging biomarkers of CSVD. Retinal microvascular abnormalities identified via either fundus photography, OCT or OCTA have so far generated the largest amount of published evidence for association with CSVD. However, larger studies employing standardized methodologies for both retinal imaging and CSVD characterization are required to definitively establish the potential impact of these technologies in future research efforts and clinical practice.”

We have modified the Discussion section of the revised manuscript (Page 39, lines 1 to 15) to more clearly summarize our findings, identify knowledge gaps in the field, and present detailed recommendations for future retinal biomarkers studies in CSVD.

Reviewer #2: Biffi et al have written an important systematic review “Retinal imaging biomarkers of cerebral small vessel disease”.

We thank Reviewer 2 for their encouraging remarks.

Please find below itemized responses to your queries and suggestions.

Addition of visual fields, electroretinography, and visual evoked potentials along with OCTA, OCT, fundus photograph have added more value to the review, however, they don’t seem to be imaging modalities. Therefore, retinal biomarkers….seem to be a better title.

In agreement with Reviewer 2’s suggestion, we modified the title of our revised manuscript to “Retinal Biomarkers of Cerebral Small Vessel Disease: a Systematic Review”. We have also modified the manuscript to more consistently refer to retinal biomarkers.

I have few minor comments.

Please expand the abbreviations in the abstract. CAA, or HTNA.

We have modified the abstract of the revised manuscript to expand on the abbreviations included.

Can you please clearly define CAA/HTNA/CADASIL/MELAS for general readers in the introduction.

We have modified the Introduction section of the revised manuscript (Page 6, lines 1 to 8) and created a dedicated table (Table 1) to provide general readers with an overview of the classification, epidemiology, and genetics of different forms of CSVD

Adding the different software used to measure retinal metrics and/or criteria for classification of CSVD in the table would add more value to the review

We modified Tables 3-7 in the revised manuscript to address this comment from Reviewer 2.

---

## [Editor Report · Decision Letter 1]

31 Mar 2022

Retinal Biomarkers of Cerebral Small Vessel Disease: a Systematic Review

PONE-D-22-00086R1

Dear Dr. Biffi,

We’re pleased to inform you that your manuscript has been judged scientifically suitable for publication and will be formally accepted for publication once it meets all outstanding technical requirements.

Kind regards,

Rayaz A Malik, MBChB, PhD

Academic Editor

PLOS ONE

Additional Editor Comments (optional):

All concerns raised have been comprehensively addressed. The revision is a much improved manuscript in an important area.
---

## [Editor Report · Acceptance letter]

4 Apr 2022

PONE-D-22-00086R1 

Retinal Biomarkers of Cerebral Small Vessel Disease: a Systematic Review 

Dear Dr. Biffi:

I'm pleased to inform you that your manuscript has been deemed suitable for publication in PLOS ONE. Congratulations! Your manuscript is now with our production department. 

Kind regards, 

on behalf of

Professor Rayaz A Malik 

Academic Editor

PLOS ONE